# Genome-wide identification of splicing QTLs in the human brain and their enrichment among schizophrenia-associated loci

Atsushi Takata[1,2], Naomichi Matsumoto[2] & Tadafumi Kato[1]

Detailed analyses of transcriptome have revealed complexity in regulation of alternative splicing (AS). These AS events often undergo modulation by genetic variants. Here we analyse RNA-sequencing data of prefrontal cortex from 206 individuals in combination with their genotypes and identify *cis*-acting splicing quantitative trait loci (sQTLs) throughout the genome. These sQTLs are enriched among exonic and H3K4me3-marked regions. Moreover, we observe significant enrichment of sQTLs among disease-associated loci identified by GWAS, especially in schizophrenia risk loci. Closer examination of each schizophrenia-associated loci revealed four regions (each encompasses *NEK4*, *FXR1*, *SNAP91* or *APOPT1*), where the index SNP in GWAS is in strong linkage disequilibrium with sQTL SNP(s), suggesting dysregulation of AS as the underlying mechanism of the association signal. Our study provides an informative resource of sQTL SNPs in the human brain, which can facilitate understanding of the genetic architecture of complex brain disorders such as schizophrenia.

[1] Laboratory for Molecular Dynamics of Mental Disorders, RIKEN Brain Science Institute, 2-1 Hirosawa, Wako-shi, Saitama 351-0198, Japan. [2] Department of Human Genetics, Yokohama City University Graduate School of Medicine, Yokohama 236-0004, Japan. Correspondence and requests for materials should be addressed to A.T. (email: atakata@brain.riken.jp) or to T.K. (email: kato@brain.riken.jp).

Alternative splicing (AS) is the process by which different splice sites in precursor messenger RNA are selected to generate multiple mRNA isoforms. AS events are often regulated in a cell type-, condition- or species-specific manner. Notably, recent studies have demonstrated that complexity of AS regulation is highest in primates[1], and that there is a distinct and more complex pattern of AS in brain tissues[2,3]. Such highly intricate regulation of AS in the human brain can play an important role in normal function and development of the central nervous system. For example, a number of genetic mutations that affect global regulation of AS or alter AS of a specific gene are known to be associated with various brain disorders[4,5]. More recently, it was reported that a subset of de novo germline mutations, whose important roles in the genetic aetiology of neuropsychiatric disorders such as autism spectrum disorders (ASDs) and schizophrenia has been established[6–10], probably contribute to the risk of ASD and schizophrenia by affecting AS[11]. In addition, dysregulation of AS is reported in multiple postmortem brain studies of ASD[12,13] and schizophrenia[14,15].

As represented by canonical splice site variants disrupting exon–intron boundaries, regulation of AS can be controlled by genetic variants. Not only variants directly changing splice site sequences, it has been demonstrated that genetic variants controlling AS events, referred to as splicing quantitative trait loci (sQTLs), spread throughout the genome. In particular, recent large-scale studies utilizing the data of RNA sequencing (RNA-seq) have successfully identified sQTLs in a genome-wide manner[3,16]. However, these studies are primarily focusing on non-neuronal tissues and thereby sQTLs in the human brain have not yet been well characterized. Although a previous microarray-based study has identified exon-specific QTLs in brain tissues, detectable AS events depend on array design and also are restricted to exon skipping. Therefore, a study utilizing RNA-seq data has a particular advantage in identifying more AS events[17].

To comprehensively detect sQTLs in the human brain, here we analyse RNA-seq data of dorsolateral prefrontal cortex (DLPFC) tissues from >200 individuals in combination with their microarray-based genotype data. After applying stringent filtering criteria, we identify a total of ~1,500 sQTL single-nucleotide polymorphisms (SNPs) that are likely to be independent of each other. By analysing characteristics of these brain sQTL SNPs, we describe functional properties of these variants and their potential roles in the genetic aetiology of human diseases, particularly in brain disorders such as schizophrenia. We also show an example how the information of sQTLs can be utilized to better understand the complex genetic architecture of human diseases and to specify promising candidates for culprit genes using the data of large-scale genome-wide association study (GWAS) for schizophrenia[18].

## Results

**Identification of *cis*-acting splicing QTLs in human brain**. We first analysed RNA-seq data of DLPFC samples (all from Brodmann area 9) from genetically homogenous 206 individuals (Supplementary Fig. 1, extracted by using the result of multidimensional scaling) without neuropsychiatric diseases or neurological insults immediately prior to death (downloaded from the CommonMind Consortium Knowledge Portal, summary statistics are available in Supplementary Table 1, see also Methods) to comprehensively identify AS events in the human brain. For this purpose we used vast-tools[13], a software package designed to identify various types of AS events, including alternative exon skipping (Alt EX), alternative usage of splice sites (Alt SS) and intron retentions (IRs). After applying quality

control filters (see Methods for details), we identified a total of 102,469 AS events in autosomes, consisting of 29,271 Alt EX, 3,310 Alt SS (of which 1,265 were at the 5′-donor site and 2,045 were at the 3′-acceptor site) and 69,888 IRs. We next analysed this list of AS in combination with quality-controlled SNP genotyping data of the same individuals using Matrix eQTL[19] to identify *cis*-acting (within ±100 kb of the AS event) sQTLs in a genome-wide manner (see Methods for details). To conservatively define sQTL SNPs, we first applied a standard correction for multiple testing implemented in Matrix eQTL (Benjamini–Hochberg procedure) to the P-values for all SNP–AS pairs and then the corrected P-values were further subjected to Bonferroni correction with the number of AS events within the ±100 kb window for each SNP. This is because a SNP with many AS events in the surrounding region should have a higher chance to show significant association (see Methods for details). After performing these procedures, we identified a total of 8,966 sQTL SNPs with the 'double-corrected' P-value <0.05. The full list of sQTL SNPs along with information of the associated AS events is available in Supplementary Data 1. Consistent with previous studies of non-neuronal tissues[3,16], when we plotted the double-corrected P-value and the distance to the nearest AS event for each SNP, we observed that variants at the proximity of AS are enriched for sQTL SNPs (Fig. 1a).

The identified 8,966 sQTL SNPs are involved in 1,595 AS events of 1,341 unique genes. When we performed a gene-set enrichment analysis of these 1,341 genes using the Database for Annotation, Visualization and Integrated Discovery[20], we found highly significant enrichment of 'SP_PIR_KEYWORDS: alternative splicing' (Benjamini-corrected $P = 8.6 \times 10^{-29}$) and 'UP_SEQ_FEATURE: splice variants' (Benjamini-corrected $P = 1.1 \times 10^{28}$), which denote genes with known splicing isoforms (Supplementary Data 2). Therefore, on the one hand, our result is compatible with the existing knowledge of genes undergo AS and, on the other hand, the list of genes regulated by sQTL SNPs identified here provides new candidates for genes with splicing isoforms, because >40% of the input genes were not included in 'SP_PIR_KEYWORDS: alternative splicing' or 'UP_SEQ_FEATURE: splice variants' (Supplementary Data 2) but in fact have detectable alternatively spliced regions.

**Functional characterization of sQTL SNPs**. We consequently attempted to functionally characterize sQTL SNPs. For this purpose, we first extracted the best sQTL SNP for each AS event ($N = 1,595$) and then performed linkage disequilibrium (LD)-based pruning (see Methods). After performing this procedure, there was a set of 1,539 sQTL SNPs that are likely to be independent of each other. Next, we performed LD-based pruning on SNPs with an uncorrected P-value > 0.05 ($N = 170,241$) with the same parameters applied to sQTL SNPs, leaving 89,367 SNPs that are unlikely to be associated with AS (we considered these as non-sQTL SNPs). From this list of non-sQTL SNPs, we generated a set of 48,068 SNPs with the distribution of minor allele frequency (MAF) matched to the set of 1,539 sQTL SNPs and used them for comparison (see Supplementary Fig. 2 and Methods).

By functionally classifying the SNPs according to the definition in SnpEff[21], we found that sQTL SNPs are significantly enriched among variants in exonic regions (that is, nonsense, readthrough, start-loss, frameshift, canonical splice site, missense, synonymous, splice region, 5′-untranslated region (UTR), 3′-UTR and non-coding exon variants; shown in warm colours in Fig. 1b) when compared with non-sQTL SNPs ($P = 8.6 \times 10^{-87}$, odds ratio (OR) = 3.84, two-tailed Fisher's exact test). By analysing enrichment of sQTL SNPs among each functional type of variants, as expected, we found significant enrichment with the

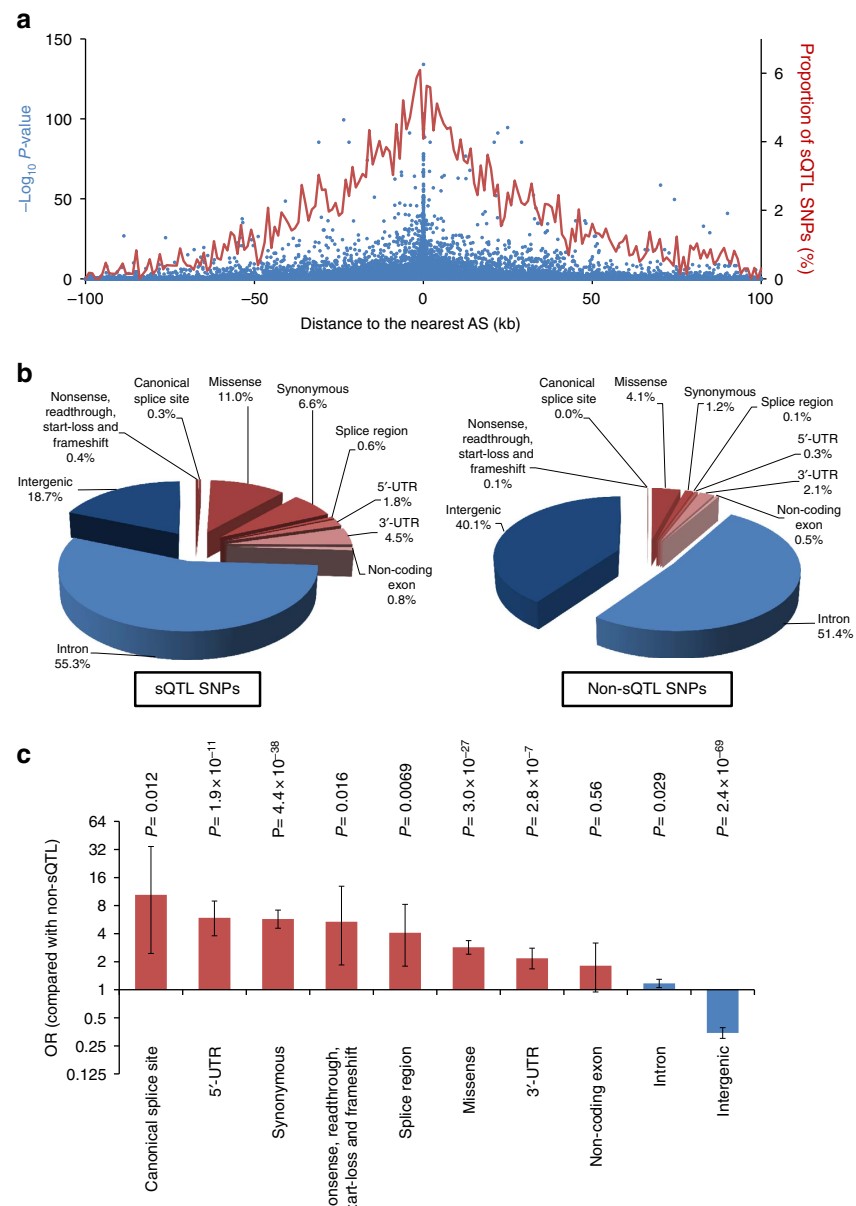

**Figure 1 | Characterization of identified sQTL SNPs.** (**a**) Each blue dot indicates a SNP plotted according to its distance to the nearest AS event and statistical significance for association with AS (–$\log_{10}$ $P$-value). Red line indicates proportion of SNPs (%) that were classified as sQTL SNPs. Proportions in each 1,000 bp window were plotted. (**b**) Pie charts indicating proportions of SNPs annotated with each functional category (nonsense, readthrough, start-loss, frameshift, canonical splice site, missense, synonymous, splice region, 5'-UTR, 3'-UTR, non-coding exon, intron and intergenic). SNPs in exonic regions (nonsense, readthrough, start-loss, frameshift, canonical splice site, missense, synonymous, splice region, 5'-UTR, 3'-UTR and non-coding exon) and SNPs in non-exonic regions (intron and intergenic) are indicated by warm and cold colours, respectively. (**c**) Enrichment analyses of sQTL SNPs in each different functional type of variants. Exonic variants are shown in red and non-exonic variants are shown in blue. $P$-values were calculated by two-tailed Fisher's exact test with Bonferroni correction according to the number of functional types analysed (that is, ten types). Bars indicate 95% confidence intervals.

highest OR among canonical splice site variants ($P = 0.012$, OR $= 10.4$, two-tailed Fisher's exact test with Bonferroni correction), followed by 5'-UTR and synonymous variants (Fig. 1c). In contrast, there was significant underrepresentation of sQTL SNPs among intergenic variants.

By manually inspecting all individual sQTL SNPs at the canonical splice sites ($N = 9$, from the full list of 8,966 SNPs before pruning), we found that 8 out of the 9 SNPs are associated with AS of the adjacent exon. The remaining one sQTL SNP (rs8873 at chr11: 58,378,424 in *ZFP91*) is at a splice site that is found in the RefSeq Genes track but not in the Ensembl Gene Predictions track of the UCSC Genome Browser

(https://genome.ucsc.edu/) (Supplementary Fig. 3) and transcripts spliced at this position (chr11: 58,378,426) were not detected in our analysis. Among the eight sQTL SNPs associated with AS of the adjacent exon, three variants are contributing to known (annotated by Ensembl Gene Predictions) AS events (Fig. 2). In the case of rs2276611 at chr2: 170,441,001 in *PPIG*, alternative splice sites are almost exclusively used depending on the alleles (Fig. 2a). Around rs3803354 at chr15: 40,856,989 in *C15orf57*, there are three different splice sites (Fig. 2b). Although the major isoform is spliced at chr15: 40,857,175 (blue arrow head in Fig. 2b), proportion of the isoform spliced at chr15: 40,856,990 (red arrow head) increases in C allele carriers in an additive

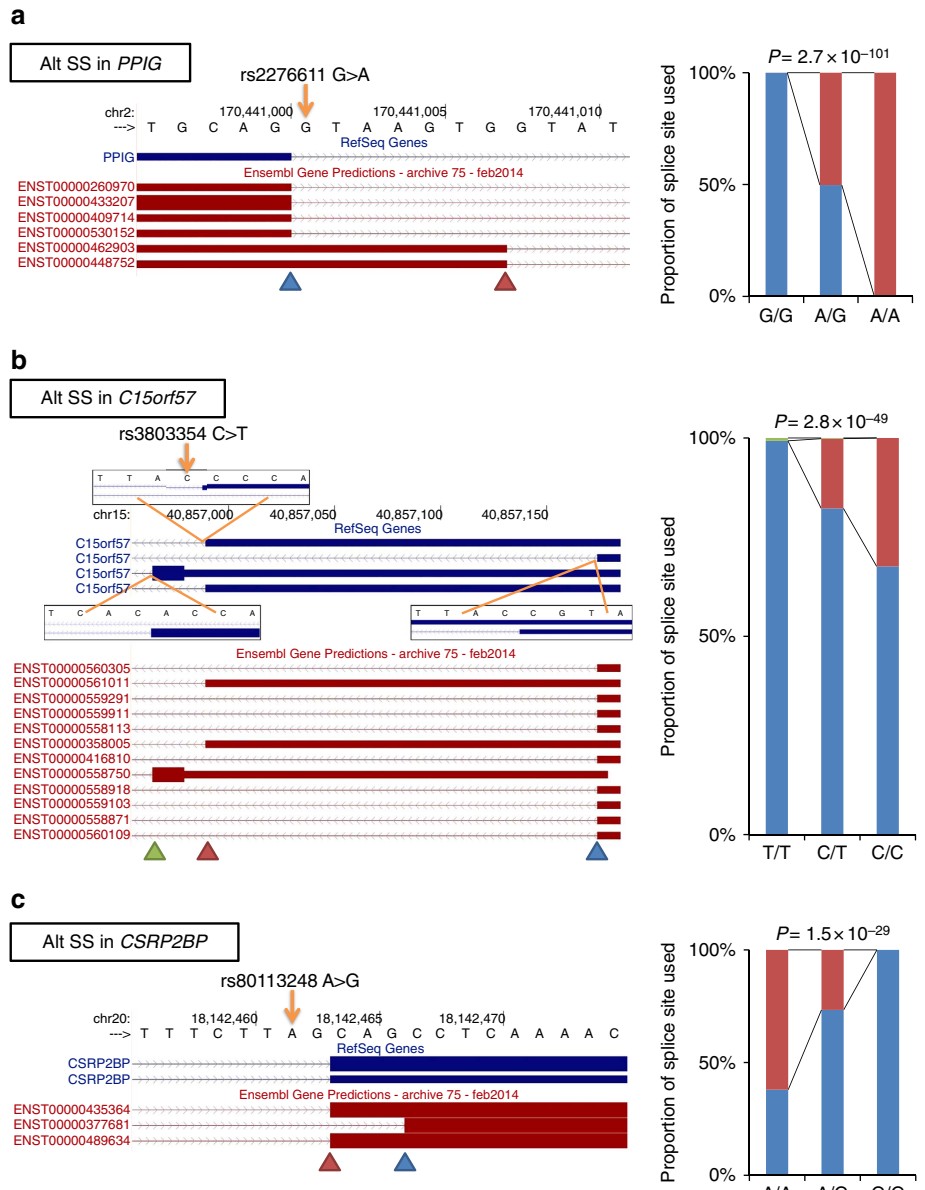

**Figure 2 | sQTL SNPs at canonical splice sites of genes with known transcript isoforms.** sQTL SNPs at the canonical splice sites of *PPIG* (**a**), *C15orf57* (**b**) and *CSRP2BP* (**c**) controlling alternative usage of splice sites. Schematic of transcript isoforms at each locus (RefSeq Genes and Ensembl Gene Predictions tracks from the UCSC Genome Browser (https://genome.ucsc.edu/) with the genomic sequences and coordinates) are shown in the left panels. Orange arrows indicate the positions of sQTL SNPs. Arrowheads indicate alternative splice sites. In **b**, detailed sequences around three differently used splice sites (chr15: 40,856,965, 40,856,990 and 40,857,175) are shown in magnified view. Proportions of alternative splice sites used are shown in the right panels. The averages among the carriers of each genotype are shown as stacked bars. The colours of stacked bars (blue, red and green) correspond to the alternative splice sites (arrowheads) in the left panels. Double-corrected *P*-values (see Methods) are indicated above the bars.

manner and also there is a minor isoform (average percent-spliced-in (PSI) < 1) spliced at chr15: 40,856,965 (green arrow head). In the case of rs80113248 at chr20: 18,142,462 in *CSRP2BP*, both the two splice sites 3 bp distant to each other (chr20: 18,142,464 and 18,142,467) are used in A allele carriers, whereas in G/G carriers the transcripts are exclusively spliced at chr20: 18,142,467 (Fig. 2c). For the other five canonical splice site sQTL SNPs at the proximity of associated AS, we also found that disruption of canonical splice site by the variant allele causes increased proportion of exon skipping or IR (Supplementary Fig. 4). Although the number of canonical splice site variants analysed in this study is small, identification of these 'positive control' variants regulating AS in an expected way could support the validity of our analyses.

**sQTL SNPs and genetic regulatory elements.** In a recent study of non-neuronal tissues, enrichment of sQTL SNPs among various regulatory elements was reported[3]. By using the data of the ENCODE project[22], we analysed whether the brain sQTL SNPs identified in this study are enriched among variants within genomic regions with the following regulatory annotations; DNase I hypersensitive sites, monomethylated histone H3 lysine 4 (H3K4me1), trimethylated histone H3 lysine 4 (H3K4me3), acetylated histone H3 lysine 9 (H3K9ac), acetylated histone H3 lysine 27 (H3K27ac) and transcription factor (TF) binding sites. We found significant enrichment of sQTL SNPs among variants within H3K4me3 marks ($P = 1.7 \times 10^{-11}$, OR = 2.10, two-tailed Fisher's exact test with Bonferroni correction) and significant depletion of these SNPs among H3K4me1 ($P = 9.0 \times 10^{-6}$,

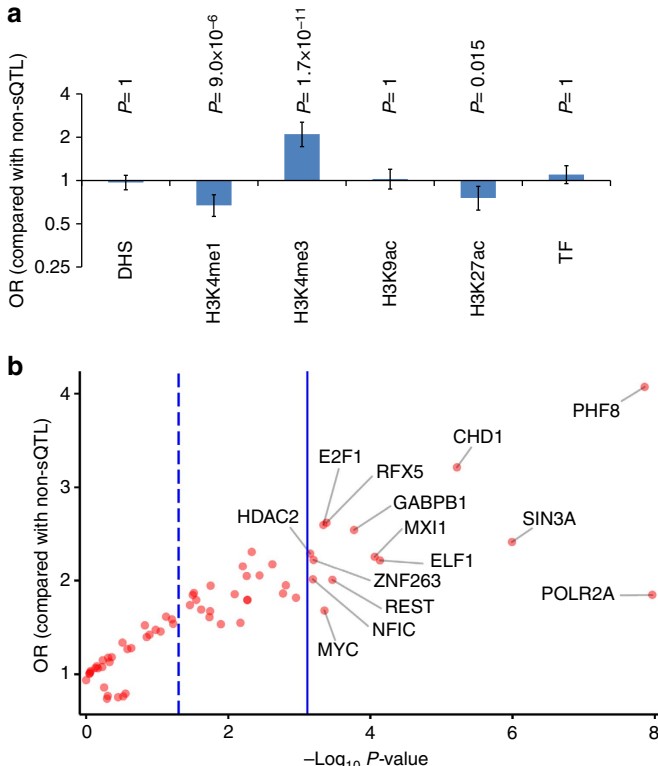

**Figure 3 | Enrichment analyses of sQTL SNPs among variants within genetic regulatory elements.** (**a**) Enrichment analysis of sQTL SNPs among variants within six types of regulatory elements (DNase I hypersensitive sites (DHS), H3K4 monomethylation marks (H3K4me1), H3K4 trimethylation marks (H3K4me3), H3K9 acetylation marks (H3K9ac), H3K27 acetylation marks (H3K27ac) and TF binding sites). P-values were calculated by two-tailed Fisher's exact test with Bonferroni correction according to the number of regulatory elements analysed (six elements). Bars indicate 95% confidence intervals. (**b**) Plots of $-\log_{10}$ P-values (x axis) and OR (y axis) obtained from enrichment analysis of sQTL SNPs among variants within binding sites for each TF. The dashed blue line indicates $P = 0.05$ and the solid blue line indicates $P = 0.05/65 = 7.7 \times 10^{-4}$ (Bonferroni-corrected P-value threshold, binding sites for a total of 65 TF were tested).

OR $= 0.67$) and H3K27ac ($P = 0.015$, OR $= 0.76$) variants (Fig. 3a). We next looked at the data of binding sites for individual TF. After performing Bonferroni correction with the number of TF subjected to our analyses (65 TF in total), significant enrichment of sQTL SNPs was observed for 14 TF (Fig. 3b and Supplementary Data 3). The most significant enrichment was observed for POLR2A-binding sites ($P = 7.1 \times 10^{-17}$, OR $= 1.85$, two-tailed Fisher's exact test with Bonferroni correction), followed by PHF8 with the highest OR ($P = 9.0 \times 10^{-7}$, OR $= 4.07$) and SIN3A ($P = 6.6 \times 10^{-5}$, OR $= 2.41$), CHD1 ($P = 0.00039$, OR $= 3.21$) and ELF1 ($P = 0.0048$, OR $= 2.22$).

**Enrichment analysis of sQTLs among disease-associated loci.** To analyse the property of sQTL SNPs in the context of their potential contribution to disease risks, we performed enrichment analyses using the data of the GWAS Catalog[23], a collection of data from GWAS for various human diseases and traits (see Methods for definition of the associated loci). When we tested whether sQTL SNPs are globally enriched among loci associated with various human diseases (defined by the Experimental Factor

Ontology (EFO)[24] term 'EFO_0000408: disease'), we found significant enrichment when compared with non-sQTL SNPs ($P = 1.7 \times 10^{-8}$, OR $= 1.33$, one-tailed Fisher's exact test). We next analysed enrichment of sQTL SNPs using the data of nine individual diseases with the largest numbers of genome-wide significantly associated SNPs in the Catalog (breast cancer, colorectal cancer, inflammatory bowel disease, multiple sclerosis, prostate cancer, psoriasis, rheumatoid arthritis, schizophrenia and type 2 diabetes), as well as four additional brain disorders (autism, Alzheimer's disease, bipolar disorder and Parkinson's disease) and two most intensively investigated non-disease traits (height and body mass index). We observed significant enrichment of sQTL SNPs among the loci associated with inflammatory bowel disease ($P = 0.0065$, OR $= 1.41$, one-tailed Fisher's exact test with Bonferroni correction), schizophrenia ($P = 0.0092$, OR $= 2.53$) and psoriasis ($P = 0.011$, OR $= 2.57$) after performing correction for multiple testing (Fig. 4a). As we found that in some cases (for example, in the case of psoriasis) the enrichment was mostly driven by variants in the major histocompatibility complex (MHC) locus when checking individual SNPs in the associated loci, we also performed enrichment analyses excluding the data of SNPs in the MHC locus. In these analyses, there was significant enrichment of sQTL SNPs among the loci associated with schizophrenia ($P = 9.9 \times 10^{-5}$, OR $= 3.72$), inflammatory bowel disease ($P = 0.0014$, OR $= 1.43$) and multiple sclerosis ($P = 0.036$, OR $= 3.71$) (Fig. 4a).

In line with the fact that the data set used in this study derives from brain tissues, diseases whose associated loci are enriched for sQTL SNPs with the highest ORs include autism, schizophrenia and multiple sclerosis, whereas enrichment among autism-associated loci was not statistically significant (Fig. 4a, analyses excluding MHC variants). Among these diseases, most statistically significant enrichment was observed for schizophrenia-associated loci ($P = 9.9 \times 10^{-5}$ after performing Bonferroni correction). We next focused on this observation and performed several confirmatory analyses to test the credibility of this result. First, we repeated the analysis using the data of well-defined 108 schizophrenia-associated loci described in the largest GWAS to date conducted by the Psychiatric Genomics Consortium[18] (PGC GWAS). This was because some of the SNPs identified by PGC GWAS were not included in the GWAS Catalog and the associated loci were defined in a more sophisticated way in PGC GWAS. With this data set, we confirmed that there was significant enrichment of sQTL SNPs among the risk loci (Fig. 4b, $P = 1.1 \times 10^{-7}$, OR $= 4.01$, one-tailed Fisher's exact test). Second, to test whether the enrichment is driven by higher proportion of exonic variants among sQTL SNPs (these variants would be more likely to be functional and thereby associated with schizophrenia regardless of their impacts on AS), we performed an analysis using the data of SNPs in non-exonic (that is, intronic and intergenic) regions (N of SNPs $= 1,139$). We found that non-exonic sQTL SNPs are significantly enriched among schizophrenia-associated loci when compared with non-sQTL SNPs in non-exonic regions (Fig. 4c, $P = 0.0030$, OR $= 2.66$, one-tailed Fisher's exact test). On the other hand, there was no statistically significant enrichment of exonic sQTL SNPs among schizophrenia risk loci when compared with exonic non-sQTL SNPs (Fig. 4c, $P = 0.36$, OR $= 1.26$), suggesting that non-exonic sQTL SNPs are particularly contributing to schizophrenia risk by their impacts on splicing regulation. Third, we performed an analysis excluding sQTL SNPs associated with IR (N of excluded SNPs $= 398$). This was because often detection of IR is more challenging than Alt EX and Alt SS, and the RNA-seq data set used in this study derives from libraries prepared by ribosomal RNA depletion (not poly-A selection; thus, premature RNA

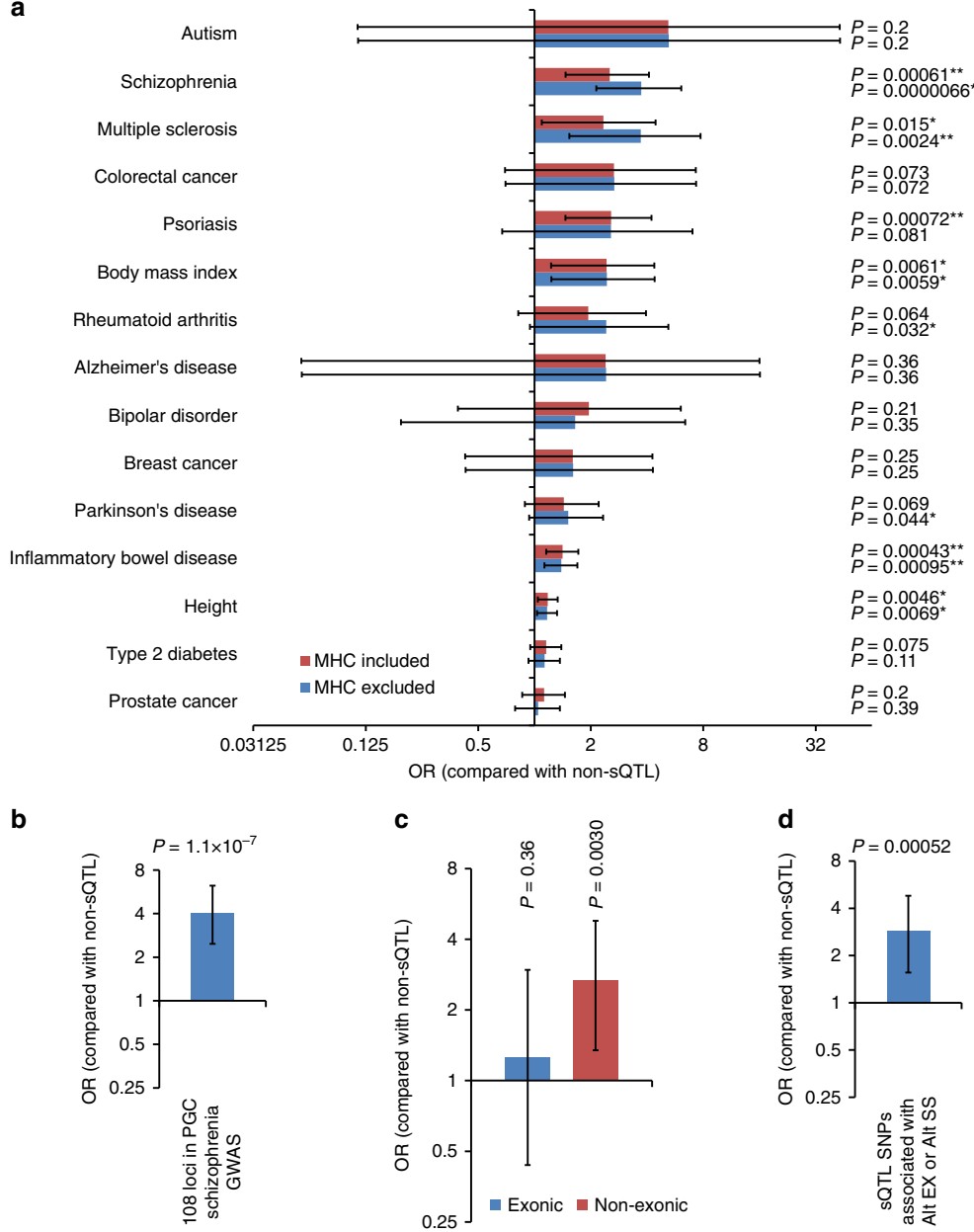

**Figure 4 | Enrichment analyses of sQTL SNPs among disease-associated loci. (a)** Results of enrichment analyses of sQTL SNPs among loci associated with 15 diseases/traits (nine diseases with the largest numbers of genome-wide significantly associated SNPs in the GWAS Catalog[23]: breast cancer, colorectal cancer, inflammatory bowel disease, multiple sclerosis, prostate cancer, psoriasis, rheumatoid arthritis, schizophrenia and type 2 diabetes; four additional brain disorder groups: autism, Alzheimer's disease, bipolar disorder, Parkinson's disease; and two most intensively investigated non-disease traits: height and body mass index). Red and blue bars indicate the results from analyses including and excluding variants in the MHC locus, respectively. Results are shown in the order of OR from the analyses excluding MHC variants. Uncorrected P-values calculated by one-tailed Fisher's exact test are shown. *$P < 0.05$ and **$P < 0.05/$ $15 = 0.0033$ (corresponding to the significance threshold considering the number of diseases/traits tested). **(b)** An enrichment analysis using the data of PGC GWAS instead of the data based on the GWAS Catalog. **(c)** Enrichment analyses dividing SNPs into exonic and non-exonic variants. **(d)** An enrichment analysis excluding sQTL SNPs associated with IRs. P-values were calculated by one-tailed Fisher's exact tests. Bars indicate 95% confidence intervals.

containing intronic regions can be to some extent included in the libraries). We found that sQTL SNPs associated with Alt EX or Alt SS are significantly enriched among schizophrenia risk loci when compared with non-sQTLs (Fig. 4d, $P = 0.00052$, $OR = 2.85$, one-tailed Fisher's exact test). Taken together, these results support credibility of the enrichment of sQTL SNPs among schizophrenia-associated loci.

**sQTLs that can be causally associated with schizophrenia.** Significant enrichment of sQTL SNPs among schizophrenia-

associated loci observed above indicates that some of these SNPs could causally contribute to the risk of schizophrenia by affecting AS. We next sought to identify plausible candidates for such sQTL SNPs. For this purpose, we utilized the data of PGC GWAS[18] and selected candidate sQTL SNPs with the following criteria: (1) in LD with an index schizophrenia-associated SNP identified in the PGC GWAS at $r^2 > 0.8$ (it is noteworthy that we considered the most significantly associated SNP with available information of LD in the 1000 Genomes March 2012 data set at each locus as the index SNP, see Methods for more details), (2) by

themselves associated with schizophrenia at the level of genome-wide significance ($P < 5 \times 10^{-8}$) and (3) included in the list of 'credible SNPs' (the sets of SNPs 99% likely to contain the causal variants; see Methods and ref. 18 for more details). We found that four schizophrenia-associated loci harbour sQTL SNPs fulfilling the selection criteria (Fig. 5). One was found on chromosome 3p21, where the index schizophrenia-associated SNP (rs2535627, $P$ for schizophrenia association in the PGC GWAS $= 4.0 \times 10^{-11}$) itself was identified as an sQTL SNP significantly associated with an Alt EX of *NEK4* (double-corrected $P$ for sQTL $= 7.8 \times 10^{-5}$) (Fig. 5a). On chromosome 3, there was another locus (3q26) with an sQTL SNP that is in strong LD with the index SNP. At this locus, rs1805564 associated with an Alt EX of *FXR1* (double-corrected $P$ for sQTL $= 0.019$) is in LD with the index SNP rs34796896 ($P$ for schizophrenia association $= 6.2 \times 10^{-11}$) at $r^2 = 0.94$ (Fig. 5b). On chromosome 6q14, an sQTL SNP rs217323 was associated with an IR of *SNAP91* (double-corrected $P$ for sQTL $= 2.1 \times 10^{-21}$) and this SNP is in LD with the index SNP rs3798869 ($P$ for schizophrenia association $= 1.2 \times 10^{-9}$) at $r^2 = 0.97$ (Fig. 5c). The last one was found on chromosome 14q32, where an sQTL SNP rs7148456 associated with an Alt EX of *APOPT1* (also known as *C14orf153*, double-corrected $P$ for sQTL $= 3.2 \times 10^{-10}$) is in LD with the index SNP rs12887734 ($P$ for schizophrenia association $= 2.3 \times 10^{-12}$) at $r^2 = 0.86$ (Fig. 5d). Identification of these SNPs suggests dysregulation of AS at these loci as plausible biological basis explaining the association signals and points to the genes whose AS is regulated by sQTL SNPs (that is, *NEK4*, *FXR1*, *SNAP91* and *APOPT1*) as promising candidates for causally associated genes among multiple genes included in each risk locus.

## Discussion

In this study, we analysed a large-scale data set of human brain transcriptome in combination with the genotyping data and identified variants controlling AS events, sQTL SNPs, in a genome-wide manner. To our knowledge, this is the first study comprehensively identifying sQTLs using RNA-seq data derived from human brain samples.

By characterizing properties of the detected sQTL SNPs, we found that these SNPs are enriched among exonic variants, including coding SNPs (Fig. 1b,c). This observation is consistent with a recently introduced notion that many of the coding variants not only define the sequence of the encoded protein but also have an impact on various regulatory functions[25,26]. We also observed that sQTL SNPs are enriched among variants within H3K4me3 marks (Fig. 3a). There is accumulating evidence that this histone mark is not only associated with transcriptional activation, but also plays a role in AS[27,28]. This process can be mediated by physical binding of spliceosome to H3K4me3 via a chromo-helicase protein CHD1 (ref. 27), whose binding sites were enriched for sQTLs (Fig. 3b). It is also known that various epigenetic marks including H3K4me3 can be locally influenced by genetic variants[29]. Therefore, some of the SNPs in H3K4me3 would alter epigenetic status and thereby act as sQTL SNPs. This possible scenario can be related to enrichment of sQTL SNPs among 5′-UTR variants, which showed the second highest OR in our analysis of various functional types of SNPs (Fig. 1c). This is because H3K4me3 marks are enriched in the 5′end of gene bodies often including 5′-UTRs[30], besides well-known enrichment at promoter regions. It would be also of note that AS of histone-modifying genes such as *KDM1A* and *EHMT2* themselves are known to play a role in global epigenetic regulation and neuronal differentiation[31]. Thus, it would be worthwhile to take this AS-chromatin feedback loop into account. In the analysis of binding sites for individual TF, we found the most significant enrichment

of sQTL SNPs among variants within binding sites for POLR2A (this protein encoding the largest subunit of RNA polymerase II is included in the list of TF in ENCODE) (Fig. 3b), which is known to be involved in AS regulation[32,33]. Strong enrichment was also observed for binding sites for various chromatin regulators such as PHF8, SIN3A and CHD1 (Fig. 3b). As partly discussed above, this observation is in concordance with their roles in regulation of AS[27,28,34,35]. Gene-set enrichment analysis of genes regulated by sQTL SNPs found enrichment of genes with known splicing isoforms, whereas it does not mean that all tested genes are involved in regulation by AS nor that all genes in the 'splicing' term could be determined by our analysis.

In the enrichment analysis of sQTL SNPs using the data of GWAS, we observed significant overrepresentation of these variants among loci associated with various human diseases, indicating roles of SNPs regulating AS in genetic disease aetiologies. This observation is in agreement with the growing evidence that the majority of SNPs identified in GWAS contribute to the disease risks through their impact on gene regulatory functions[36,37]. Specifically, we found that sQTL SNPs identified in this study using the data of human brains are strongly enriched among schizophrenia-associated loci. Besides SNPs controlling gene-level expression (eQTL) or DNA methylation (mQTL or meQTL), whose contribution to the schizophrenia risk has been demonstrated in recent studies[38–40], our results indicate that sQTL SNPs, which are in most cases not overlap with gene-level eQTLs[41,42], can explain an additional part of the genetic architecture of schizophrenia.

By utilizing the list of sQTL SNPs, we could specify four promising candidate disease susceptibility genes for schizophrenia (that is, *NEK4*, *FXR1*, *SNAP91* and *APOPT1*), whose AS are regulated by sQTL SNPs in strong LD with the index SNPs identified in the PGC GWAS. *NEK4* encodes a member of never-in-mitosis A kinase that regulates cell cycle and response to double-stranded DNA damage[43]. It is of note that this gene is most highly expressed in the brain among multiple adult human tissues[44] and plays a key role in stabilization of neuronal cilia[44], whose contribution to various neural functions including nervous system development and adult neurogenesis[45], as well as possible involvement in the pathophysiology of schizophrenia[46,47], have been reported. *FXR1* encodes a homologue of fragile-X mental retardation protein (FMRP) that is responsible for fragile X syndrome and the encoded protein (fragile X mental retardation syndrome-related protein 1) is known to interact with FMRP[48,49]. Recent large-scale genetic studies have consistently indicated involvement of FMRP targets in the genetic architectures of schizophrenia[9,50] and ASD[10,51], indicating this gene as a particularly good candidate disease-associated gene. *SNAP91* encodes the clathrin-associated protein AP180. AP180 is enriched in the presynaptic terminal of neurons[52] and play an essential role in synaptic neurotransmission[53,54]. AP180 KO mice show excitatory/inhibitory imbalance[53], which has been reported in patients and animal models of neuropsychiatric disorders including schizophrenia[55]. *APOPT1* encodes a mitochondrial protein that induces apoptotic cell death[56]. Causal contribution of this gene in cavitating leukoencephalopathy[57], a rare brain disorder, as well as accumulating evidence, suggesting involvement of mitochondrial dysfunction in neuropsychiatric disorders[58,59], imply a potential role of *APOPT1* in the pathogenesis of schizophrenia.

Considering several limitations of this study, first, although the sample size in this study is substantial ($N = 206$), it would not be sufficient to confidently identify all brain sQTL SNPs. Second, in this study we could only analyse the data of adult brain tissues from the single brain region (DLPFC). Analyses of sQTLs using large-scale data sets with higher spatial and temporal resolutions

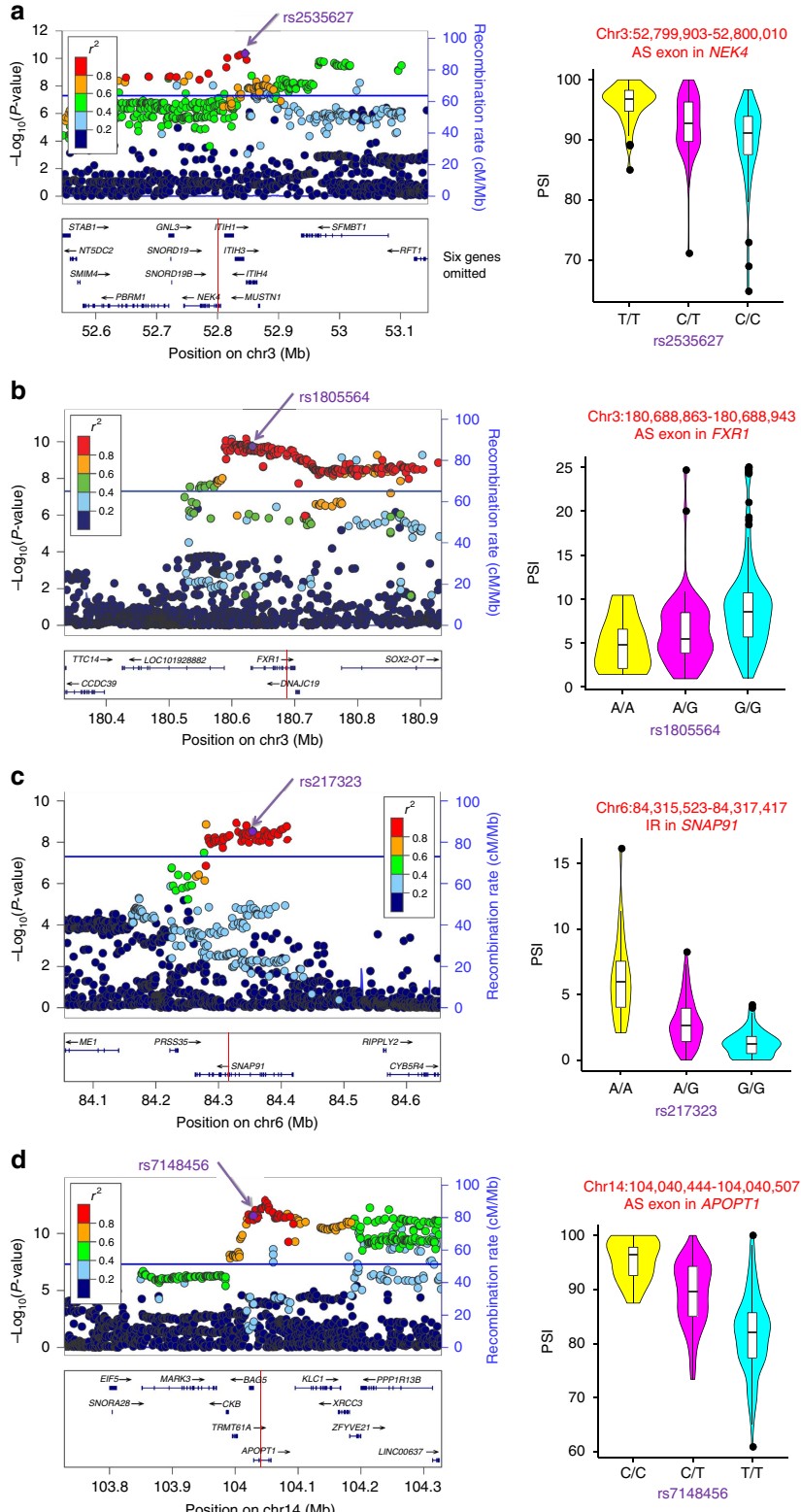

**Figure 5 | Utilization of sQTLs to localize candidate susceptibility genes for schizophrenia.** Local plots of the results of the PGC GWAS[18] (left panels) and violin plots of PSI of AS in each genotype (right panels) for four loci encompassing AS of *NEK4* (**a**), *FXR1* (**b**), *SNAP91* (**c**) and *APOPT1* (**d**), which are controlled by sQTL SNPs in strong LD ($r^2 > 0.8$) with the index SNPs in the GWAS. Local plot figures in the left panels were generated by LocusZoom[65]. Each circle indicates a SNP that are colour-coded according to their LD ($r^2$) with the sQTL SNP (indicated by purple arrows). The statistical strength of the association (–$\log_{10}$ P-values) and the recombination rate are double-plotted on the y axis. Blue horizontal lines indicate the genome-wide significance threshold ($P = 5 \times 10^{-8}$). Genes in the UCSC Genome Browser (https://genome.ucsc.edu/) are shown in the panels below the local plots. Red lines indicate the positions of the associated AS events. Violin plots in the right panels show distributions of PSI in each genotype. The overlaid boxplots indicate the median (horizontal black lines) and interquartile range (IQR; white boxes). Outliers are shown as black dots.

will provide a further informative data resource, especially in the context of identification of genes and variants associated with a disease attributable to deficits in the specific brain region(s) and/or at the particular time period(s). Third, in this study we only show statistically significant association between SNP and AS, and have not experimentally validated impact of sQTL SNPs on AS, whereas often it is very difficult to determine whether an sQTL SNP associated with AS directly regulates splicing or just tags a functional variant, which is not investigated here (such as a rare splice region variant).

In summary, we in this study comprehensively identified SNPs regulating AS events in the human brain, described the characteristics of these sQTL SNPs and demonstrated that the list of brain sQTL SNPs can be used to identify plausible candidate genes/variants causally associated with schizophreni and will also be useful to generate animal models. Our results provide a new insight into the genetic architecture of schizophrenia. By integrating various data resources (for example, sQTLs, eQTLs, mQTLs and more), we will obtain a more detailed picture of the genomic landscape of complex brain disorders.

## Methods

**RNA-seq data of DLPFC.** RNA-seq data (BAM files) of DLPFC from individuals without neuropsychiatric diseases or neurological insults immediately before death (N of individuals = 285) were downloaded from 'Raw' directory of the CommonMind Consortium Knowledge Portal (https://www.synapse.org/#!Synapse:syn4923029) using Synapse Python Client (http://python-docs.synapse.org/index.html). The data set was generated as a part of the CommonMind Consortium supported by funding from Takeda Pharmaceuticals Company Limited, F. Hoffman-La Roche Ltd and NIH grants R01MH085542, R01MH093725, P50MH066392, P50MH080405, R01MH097276, RO1-MH-075916, P50M096891, P50MH084053S1, R37MH057881 and R37MH057881S1, HHSN271201300031C, AG02219, AG05138 and MH06692. Brain tissue for the study was obtained from the following brain bank collections: the Mount Sinai NIH Brain and Tissue Repository, the University of Pennsylvania Alzheimer's Disease Core Center, the University of Pittsburgh NeuroBioBank and Brain and Tissue Repositories, and the NIMH Human Brain Collection Core. CMC Leadership: Pamela Sklar, Joseph Buxbaum (Icahn School of Medicine at Mount Sinai), Bernie Devlin, David Lewis (University of Pittsburgh), Raquel Gur, Chang-Gyu Hahn (University of Pennsylvania), Keisuke Hirai, Hiroyoshi Toyoshiba (Takeda Pharmaceuticals Company Limited), Enrico Domenici, Laurent Essioux (F. Hoffman-La Roche Ltd), Lara Mangravite, Mette Peters (Sage Bionetworks), Thomas Lehner and Barbara Lipska (NIMH). Detailed procedures for tissue collection, sample preparation, RNA-seq and data processing are available in the Consortium's wiki page (https://www.synapse.org/#!Synapse:syn2759792/wiki/69613). Briefly, ribosomal RNA was depleted from about 1 µg of total RNA using Ribozero Magnetic Gold kit (Illumina, San Diego, CA). The sequencing library was prepared using the TruSeq RNA Sample Preparation Kit v2 (Illumina). Sequencing was performed by using HiSeq2500 (Illumina). As the sequencing libraries are prepared by using rRNA depletion procedures, the RNA-seq data should contain the information from total RNA including non-coding RNA and precursor mRNA. Downloaded BAM files for mapped and unmapped reads from each individual were merged by using SAMtools[60]. Merged BAM files were converted into the fastq format using bam2fastq (https://gsl.hudsonalpha.org/information/software/bam2fastq).

**SNP genotyping data.** Quality-controlled genotyping data (SNPs with zero alternate alleles, genotyping call rate < 0.98 or Hardy–Weinberg P-value < 5 × 10^−5 and individuals with genotyping call rate < 0.90 were removed) were downloaded from 'QCd' directory of the CommonMind Consortium Knowledge Portal (https://www.synapse.org/#!Synapse:syn4551740). Genotyping was performed by using Infinium HumanOmniExpressExome v1.1 DNA Analysis Kit (Illumina). With these genotype data, we performed multidimensional scaling using PLINK[61]. As expected, the first dimension (the $x$ axis of Supplementary Fig. 1) represents ethnicities of the participants. We extracted the data of Caucasians included in the single largest cluster indicated by the red box in Supplementary Fig. 1 (N of individuals = 206, summary statistics for these individuals are available in Supplementary Table 1). After excluding SNPs with MAF < 1% among these 206 individuals with a homogeneous genetic background, there were 607,993 autosomal SNPs. Of these SNPs, we extracted 313,906 SNPs that are within ± 100 kb of any of the identified AS events and used them in the analysis of sQTL SNPs.

**Comprehensive detection of AS events.** Comprehensive detection of AS events was performed by using vast-tools (version 0.2.1)[13]. We first mapped the reads in the fastq files generated above onto the reference human genome (hg19) using the

'align' module of vast-tools with default parameters. Next, the results were merged into a single file containing PSI of each AS event in each individual using the 'combine' module of vast-tools. By using the quality scores in the combined file (Column 8), we first excluded AS events whose Score 1 (read coverage based on actual reads) and Score 2 (read coverage based on corrected reads) in Column 8 did not meet the minimum threshold (mapped reads ≥ 10, in principle) in > 20% of the individuals. We next excluded AS events whose PSI was 0 or 100% in > 90% of the individuals. After performing these procedures, there were a total of 102,469 AS events. According to the predefined types of AS in vast-tools[13], these were classified into Alt EX, Alt SS and IRs.

**Identification of sQTL SNPs.** Correlation between genotypes and PSI of AS was analysed by using Matrix eQTL[19] with the additive linear model. To control potential confounding factors, the following parameters were included in the analysis as covariates; gender, age of death, research institute where the samples were collected (Mount Sinai, Pennsylvania or Pittsburg), post-mortem interval, brain pH, RNA integrity number and sequencing library batch. We considered all AS-SNP pairs when the distance between AS and SNP is less than 100 kb. This ± 100 kb window was determined by referring previous studies reporting that sQTL SNPs are particularly enriched among the proximal regions[16,41,62]. When there are multiple AS events within the ± 100 kb window around a SNP, we used the smallest P-value to define sQTL SNPs. The smallest P-value for each SNP was then subjected to Bonferroni correction with the number of AS within the ± 100 kb counted by window function of BEDtools[63]. This was because a SNP with a large number of AS in the window should have higher chance to show significant association.

**Gene-set enrichment analysis of genes regulated by sQTL SNPs.** A gene-set enrichment analysis of genes with AS regulated by sQTL SNPs was performed by using the Database for Annotation, Visualization and Integrated Discovery[20] with default parameters. In total, there were 1,341 unique genes with AS regulated by sQTL SNPs. The input genes can be found in Supplementary Data 1.

**sQTL and non-sQTL data sets for comparison.** To generate a set of sQTL SNPs probably contributing to AS regulation independently of each other, we first extracted the best sQTL SNP for each AS event (N of SNPs = 1,595). We next performed LD-based pruning of these 1,595 SNPs using –indep-pairwise function of PLINK[61] with the following parameters: window size in SNPs = 50, the number of SNPs to shift the window at each step = 5 and the $r^2$ threshold = 0.5. For this analysis, the 1000 Genomes Project[64] March 2012 EUR (Europeans) data set downloaded as a part of the LocusZoom[65] package was used as the reference. After performing LD-based pruning, there were a total of 1,539 sQTL SNPs. To generate a control data set of non-sQTL SNPs, we first extracted SNPs for which the smallest uncorrected P-value was larger than 0.05 (N of SNPs = 170,241). We then performed LD-based pruning with the same parameters and the reference 1000 Genomes data set used for sQTL SNPs and generated a set of 89,367 SNPs, which are unlikely to be associated with AS and not strongly dependent of each other (non-sQTL SNPs). We then stratified these non-sQTL SNPs into 2% MAF bins and extracted 48,068 SNPs with the distribution of MAF matched to the set of 1,539 sQTL SNPs (Supplementary Fig. 2). We used these sets of 1,539 sQTL SNPs and 48,068 non-sQTL SNPs in the downstream analyses to characterize the properties of sQTL SNPs.

**Functional annotation of sQTL and non-sQTL SNPs.** We functionally annotated 1,539 sQTL and 48,068 non-sQTL SNPs by using SnpEff[21]. Information of SnpEff annotation was collected by using MyVariant.info (http://myvariant.info/) and Variant Effect Predictor (VeP)[66]. According to these annotations, SNPs were classified into the following categories: nonsense, readthrough, start-loss, frameshift, canonical splice site, missense, synonymous, splice region, 5′-UTR, 3′-UTR, non-coding exon, intron and intergenic variants. Splice region variants were defined as variants either within 1–3 bases of the exon or 3–8 bases of the intron from the splice site[21]. When a SNP was annotated with multiple functional types, we assigned the SNP to the functional class probably having the highest impact (that is, the leftmost one among the functional categories described above). We considered nonsense, readthrough, start-loss, frameshift, canonical splice site, missense, synonymous, splice region, 5′-UTR, 3′-UTR and non-coding exon variants as exonic SNPs, and intron and intergenic variants as non-exonic SNPs. Enrichment analyses of sQTL SNPs according to their functionalities were performed by two-tailed Fisher's exact test with the following 2 × 2 table: columns; sQTL SNPs and non-sQTL SNPs, rows; SNPs 'assigned' and 'not assigned' to the particular functional class. For enrichment analysis of each functional class of variants, we performed Bonferroni correction according to the number of functional types subjected to the analysis (ten types: canonical splice site, the other loss-of-function, missense, synonymous, splice region, 5′-UTR, 3′-UTR, non-coding exon, intron and intergenic variants).

**Enrichment analyses of sQTL SNPs among regulatory elements.** Annotation files for DNase I hypersensitive sites, H3K4me1, H3K4me3, acetylated histone H3

lysine 9, H3K27ac and TF-binding sites were downloaded from the ENCODE portal (https://www.encodeproject.org/data/annotations/, accessed January 2016, the data from Roadmap Epigenomics Consortium[67] were also integrated to these data sets). We analysed whether a SNP is included in each regulatory element using BEDtools[63]. Enrichment analyses of sQTL SNPs among regulatory elements were performed by two-tailed Fisher's exact test with the following $2 \times 2$ table: columns; sQTL SNPs and non-sQTL SNPs, rows; SNPs within and not within the regulatory element. In the enrichment analyses of binding sites for individual TF, we excluded TF for which the number of records (each record is $\sim 150$ bp genomic region) in the annotation file was smaller than 50,000. There were a total of 65 TF with 50,000 or more records of binding sites in the bed file downloaded from the ENCODE portal. According to this number we applied Bonferroni correction.

**Enrichment analyses of sQTLs among disease-associated loci.** The list of SNPs associated with various human traits was downloaded from the GWAS Catalog[23] (http://www.ebi.ac.uk/gwas, the gwas_catalog_v1.0.1 file, accessed June 2016). We included SNPs with genome-wide significant association ($P < 5 \times 10^{-8}$) in our analyses. An associated genomic locus for each SNP was defined as the genomic region containing SNPs in LD with the index SNP at $r^2 > 0.6$. SNPs in LD with the index SNP were identified by using PLINK[61] with the 1000 Genomes Project March 2012 EUR data set. Next, we analysed whether each SNP falls within the disease-associated loci using the BEDtools[63]. SNPs associated with human diseases were extracted by using the EFO[24] ID tags (MAPPED_TRAIT_URI column of the GWAS catalog file). We considered SNPs associated with any of the child terms of 'EFO_0000408: disease' as disease-associated SNPs. Information of child terms of 'EFO_0000408: disease' was collected by using the ontoCAT package[68] of R. We evaluate whether there is enrichment of sQTL SNPs among associated loci by one-tailed Fisher's exact test with the following $2 \times 2$ table: columns; sQTL SNPs and non-sQTL SNPs, rows; SNPs within and not within the disease-associated loci. We performed these analyses for the following diseases/traits: (1) all human diseases (EFO_0000408: disease); (2) nine individual diseases with the largest numbers of genome-wide significantly associated SNPs in the GWAS Catalog[23] ($N$ of SNPs $\geq 80$): breast cancer, colorectal cancer, inflammatory bowel disease (including Crohn's disease and ulcerative colitis), multiple sclerosis, prostate cancer, psoriasis, rheumatoid arthritis, schizophrenia and type 2 diabetes; (3) four additional brain disorders: autism, Alzheimer's disease, bipolar disorder and Parkinson's disease; and (4) two most intensively investigated non-disease traits: height and body mass index. For analyses excluding SNPs in the MHC locus, we did not use the information of SNPs in chr6:28,477,797–33,448,354 (hg19; based on the definition by The Genome Reference Consortium http://www.ncbi.nlm.nih. gov/projects/genome/assembly/grc/region.cgi?name=MHC&asm=GRCh37).

**Confirmatory analyses for sQTLs in schizophrenia risk loci.** A confirmatory enrichment analysis using the data of 108 loci defined in PGC GWAS[18] was performed using the data downloaded from the PGC portal (https:// www.med.unc.edu/pgc/files/resultfiles/scz2.regions.zip). Enrichment analyses excluding exonic variants were performed by extracting the data of intronic and intergenic variants according to the SnpEff[21] annotations described above (see 'Functional Annotation of sQTLs and Non-sQTL SNPs' section). An analysis excluding sQTL SNPs associated with IR was performed by excluding 398 sQTL SNPs whose most significantly associated AS was IR. Enrichment analyses of sQTL SNPs among schizophrenia-associated loci using the data of PGC GWAS, excluding the data of exonic variants or sQTL SNPs associated with IR were performed by one-tailed Fisher's exact test.

**Identification of sQTL SNPs in strong LD with the index SNP.** The full result of the PGC schizophrenia GWAS[18] (https://www.med.unc.edu/pgc/files/resultfiles/ scz2.snp.results.txt.gz) and the data of credible causal sets of SNPs (sets of SNPs that were 99% likely to contain the causal variants[18]; these sets were defined for each schizophrenia-associated locus https://www.med.unc.edu/pgc/files/ resultfiles/pgc.scz2.credible.SNPs.zip) were downloaded from the PGC portal (http://www.med.unc.edu/pgc/downloads). By using these data sets, we extracted sQTL SNPs that are: (1) in LD with an index schizophrenia-associated SNP identified in the PGC GWAS at $r^2 > 0.8$, (2) by themselves associated with schizophrenia at the level of genome-wide significance ($P < 5 \times 10^{-8}$) and (3) included in the list of 'credible SNPs' described above. In total, we found sQTL SNPs satisfying these criteria in four independent loci. In two instances, information of LD in the 1000 Genomes March 2012 EUR data set was not available for the index SNPs described in Supplementary Table 2 of ref. 18 (chr3_180594593_I and chr6_84280274_D). In these cases, we considered the most significantly associated SNP with available information of LD in each locus as the index SNP (rs34796896 for chr3_180594593_I and rs3798869 for chr6_84280274_D). Regional visualization of the PGC GWAS result (the scz2.snp.results.txt.gz file) with information of sQTL SNPs and associated AS was performed by using LocusZoom[65] based on the 1000 Genomes March 2012 EUR data set. SNPs not included in this reference data set were not displayed in the figure. LD ($r^2$) between the index SNP and sQTL SNP was computed by using PLINK[61] with the same 1000 Genomes March 2012 EUR data set.

**Data availability.** The mapped RNA-seq data (BAM files) that support the findings of this study are available in CommonMind Consortium Knowledge Portal (https://www.synapse.org/#!Synapse:syn4923029) upon authentication by the Consortium.

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

## Acknowledgements

This work was supported by JSPS KAKENHI Grant Number JP 16H06254, the Strategic Research Program for Brain Sciences from Japan Agency for Medical Research and development (AMED) and grants to the Laboratory for Molecular Dynamics of Mental Disorders, RIKEN BSI.

## Author contributions

A.T. designed the study, performed the analyses and wrote the paper. N.M. and T.K. supervised the study and contributed to the interpretation of the results.

## Additional information

**Competing financial interests:** T.K. received a research grant from Takeda Pharmaceuticals Company Limited outside of this work. The remaining authors declare no competing financial interests.

