## [Peer Review File · Nature Communications]

Reviewers' comments:

Reviewer #1 (Remarks to the Author):

This is an interesting and potentially important manuscript that studies genome-wide splicing QTLs in postmortem human brain that are enriched in schizophrenia-associated loci from the Psychiatric Genomic Consortium. Postmortem brains and the RNA sequencing and genomic data come from Common Mind, a consortium of brains from several brain collections including Mt Sinai, Pittsburgh and the University of Pennsylvania. While these different sources of tissue are covaried for in the data analyses, it would improve the manuscript if some summary data was provided including numbers from each collection that are actually used in the data, ages, RNA quality, sex and toxicology. While I doubt that any of these would adversely impact the major findings, there is no good reason to make it difficult to access these essentials of methodology. Regardless, the results are novel, convincing and are likely to influence thinking in the schizophrenia research field. The statistics seem solid, but that will require a statistician to evaluate.

Reviewer #2 (Remarks to the Author):

Takata et al. analyzed RNA sequencing data of dorsolateral prefrontal cortex from 206 individuals along with their genotypes and identify cis-acting splicing quantitative trait loci (sQTLs) SNPs throughout the genome. They also characterized their functional properties and found that sQTL SNPs are enriched among exonic regions as well as genomic regions marked by trimethylated histone H3 lysine 4.

They then analyzed the potential contribution of sQTLs to human diseases using GWAS data and observed that sQTL SNPs are significantly enriched among genomic loci associated with various human diseases including schizophrenia.

Further analysis at each schizophrenia-associated locus, identified four regions in which the index SNP in the large-scale GWAS is in strong linkage disequilibrium with sQTL SNPs. The authors propose that in these loci, dysregulation of AS by the sQTL SNP could serve as a plausible biological basis of the association signal and indicate four genes (NEK4, FXR1, SNAP91 and APOPT1) -whose AS were controlled by sQTL SNPs- as promising candidate genes.

This is a well-done, informative study and provides a useful resource that can be used in conjunction with human genetics studies to identify candidate risk genes in psychiatric and neurological disorders. I have the following comments for further improvement of the manuscript.

1. The finding of enrichment/depletion of sQTLs in histone methylation sites is interesting. It would be helpful if the authors elaborate more on this finding and discuss the potential mechanisms underlying the relation between histone methylation and alternative splicing since histone methylation is usually associated with transcriptional activation/repression [i.e., does it involve transcription from alternative start sites or alteration of spliceosome complexes? See for example: Histone methylation, alternative splicing and neuronal differentiation. Fiszbein A, et al. Neurogenesis (Austin). 2016]
2. page 5: enrichment in splicing GO does not mean that all tested genes are involved nor that the involvement of other individual genes in the list could be determined by this method. Please revise.
3. sQTL snps were identified by their locations near to splicing sites. It is therefore not surprising to see enrichment at splice sites. Does the enrichment in 5UTRs suggest mechanism as in comment #1? It would be helpful to discuss further the functional implications of the enrichment patterns.

REVIEWERS' COMMENTS:

Reviewer #1 (Remarks to the Author):

The authors have answered all relevant questions/requests in the revised manuscript.

Reviewer #2 (Remarks to the Author):

I am satisfied with the revisions made by the authors in response to my critiques.

Response to Reviewers' comments:

Response to the comments by Reviewer #1:

This is an interesting and potentially important manuscript that studies genome-wide splicing QTLs in postmortem human brain that are enriched in schizophrenia-associated loci from the Psychiatric Genomic Consortium. Postmortem brains and the RNA sequencing and genomic data come from Common Mind, a consortium of brains from several brain collections including Mt Sinai, Pittsburgh and the University of Pennsylvania. While these different sources of tissue are covaried for in the data analyses, it would improve the manuscript if some summary data was provided included numbers from each collection that are actually used in the data, ages, RNA quality, sex and toxicology. While I doubt that any of these would adversely impact the major findings, there is no good reason to make it difficult to access these essentials of methodology. Regardless, the results are novel, convincing and are likely to influence thinking in the schizophrenia research field. The statistics seem solid, but that will require a statistician to evaluate.

We are gratified by the Reviewer's interest in our manuscript and appreciation of its potential importance. We thank the Reviewer for pointing out an important issue regarding various confounding factors, which is essential in post-mortem brain studies. Indeed, while summary statistics of several factors from the whole dataset were provided in **Supplementary Table 1** of the previous version of our manuscript, there was no data stratified by the participating institutes. We therefore updated **Supplementary Table 1** as follows.

Variables	Whole dataset (N=206)	Mt. Sinai (N=108)	Pennsylvania (N=25)	Pittsburg (N=73)
Age of death (years, average \pm SD) ^a	64.6 \pm 19.4	75.1 \pm 16.4	68.5 \pm 16.1	49.5 \pm 13.6
Post-mortem interval (hours, average \pm SD)	14.5 \pm 7.8	11.9 \pm 8.0	13.5 \pm 7.3	19.2 \pm 5.2
pH (average \pm SD)	6.6 \pm 0.3	6.5 \pm 0.3	6.4 \pm 0.3	6.7 \pm 0.2
RNA integrity number (average \pm SD)	7.8 \pm 0.9	7.4 \pm 0.9	7.5 \pm 0.7	8.5 \pm 0.4
Mapped number of reads (millions, average \pm SD)	78.6 \pm 20.6	76.9 \pm 17.7	87.1 \pm 19.9	78.3 \pm 24.2
Total number of reads (millions, average \pm SD)	87.4 \pm 24.3	85.5 \pm 18.5	95.1 \pm 21.3	87.7 \pm 31.4
Percent aligned (average \pm SD)	90.0 \pm 4.6	89.8 \pm 5.1	91.5 \pm 2.9	89.9 \pm 4.3
Gender (male: female)	119: 87	54: 54	13: 12	52: 21

^aAge of death \geq 90 was considered as 90 due to limited information in the original data source.

Response to the comments by Reviewer #2:

Takata et al. analyzed RNA sequencing data of dorsolateral prefrontal cortex from 206 individuals along with their genotypes and identify cis-acting splicing quantitative trait loci (sQTLs) SNPs throughout the genome. They also characterized their functional properties and found that sQTL SNPs are enriched among exonic regions as well as genomic regions marked by trimethylated histone H3 lysine 4.

They then analyzed the potential contribution of sQTLs to human diseases using GWAS data and observed that sQTL SNPs are significantly enriched among genomic loci associated with various human diseases including schizophrenia.

Further analysis at each schizophrenia-associated locus, identified four regions in which the index SNP in the large-scale GWAS is in strong linkage disequilibrium with sQTL SNPs. The authors propose that in these loci, dysregulation of AS by the sQTL SNP could serve as a plausible biological basis of the association signal and indicate four genes (NEK4, FXR1, SNAP91 and APOPT1) -whose AS were controlled by sQTL SNPs- as promising candidates genes.

This is a well-done, informative study and provides a useful resource that can be used in conjunction with human genetics studies to identify candidate risk genes in psychiatric and neurological disorders. I have the following comments for further improvement of the manuscript.

We thank the Reviewer for his/her careful reading of our manuscript, appreciation of the significance of our work and constructive comments for improvement of our paper.

1.The finding of enrichment/depletion of sQTLs in histone methylation sites is interesting. It would be helpful if the authors elaborate more on this finding and discuss the potential mechanisms underlying the relation between histone methylation and alternative splicing since histone methylation is usually associated with transcriptional activation/repression [i.e., does it involve transcription from alternative start sites or alteration of spliceosome complexes? See for example: Histone methylation, alternative splicing and neuronal differentiation. Fiszbein A, et al. Neurogenesis (Austin). 2016]

We appreciate the Reviewer's insightful comment. We agree that it is very important to consider on possible mechanisms underlying the observed enrichment of sQTLs in H3K4me3 peaks. As indicated by the Reviewer, this issue is also related to the comment #3. We therefore revised **Discussion** in conjunction with our response to the comment #3 as follows;

(Page 12, line 298)

There is accumulating evidence that this histone mark is not only associated with transcriptional activation, but also plays a role in alternative splicing^{27, 28}. This process can be mediated by physical binding of spliceosome to H3K4me3 via a chromo-helicase protein CHD1²⁷, whose binding sites were enriched for sQTLs (Figure 3b). It is also known that various epigenetic marks including H3K4me3 can be locally influenced by genetic variants²⁹. Therefore, some of the SNPs in H3K4me3 would alter epigenetic status and thereby act as sQTL SNPs. This possible scenario can be related to enrichment of sQTL SNPs among 5' UTR variants, which showed the second highest OR in our analysis of various functional types of SNPs (Figure 1c). This is because H3K4me3 marks are enriched in 5' end of gene bodies often including 5' UTRs³⁰, besides well-known enrichment at promoter regions. It would be also of note that AS of histone modifier genes such as KDM1A and EHMT2 themselves are known to play a role in global epigenetic regulation and neuronal differentiation³¹. Thus, it would be worthwhile to take this AS-chromatin feedback loop into account.

2. page 5: enrichment in splicing GO does not mean that all tested genes are involved nor that the involvement of other individual genes in the list could be determined by this method. Please revise.

We appreciate the Reviewer for clarifying the correct interpretation of the result of GO enrichment analysis. We revised **Discussion** and **Results** as follows;

(Page 13, line 318)

Gene-set enrichment analysis of genes regulated by sQTL SNPs found enrichment of genes with known splicing isoforms, whereas it does not mean that all tested genes are involved in regulation by AS nor that all genes in the “splicing” term could be determined by our analysis.

(Page 5, line 117)

Before:

Therefore on one hand, our result is consistent with the existing knowledge of genes undergo AS, and on the other hand, the list of genes regulated by sQTL SNPs identified here provides new information because more than 40 % of the input genes were not included in “SP_PIR_KEYWORDS: alternative splicing” or “UP_SEQ_FEATURE: splice variants” (Supplementary Table 3) but actually have detectable alternatively spliced regions.

After:

Therefore on one hand, our result is compatible with the existing knowledge of genes

*undergo AS, and on the other hand, the list of genes regulated by sQTL SNPs identified here provide new candidates for genes with splicing isoforms because more than 40 % of the input genes were not included in “SP_PIR_KEYWORDS: alternative splicing” or “UP_SEQ_FEATURE: splice variants” (**Supplementary Table 3**) but have detectable alternatively spliced regions.*

3. sQTL snps were identified by their locations near to splicing sites. It is therefore not surprising to see enrichment at splice sites. Does the enrichment in 5UTRs suggest mechanism as in comment #1? It would be helpful to discuss further the functional implications of the enrichment patterns.

We appreciate this valuable comment along with the comment #1. We revised **Discussion** as described above in our response to the comment #1.